# Factors Associated with Telehealth Utilization among Older African Americans in South Los Angeles during the COVID-19 Pandemic

**DOI:** 10.3390/ijerph20032675

**Published:** 2023-02-02

**Authors:** Tavonia Ekwegh, Sharon Cobb, Edward K. Adinkrah, Roberto Vargas, Lucy W. Kibe, Humberto Sanchez, Joe Waller, Hoorolnesa Ameli, Mohsen Bazargan

**Affiliations:** 1Mervyn M. Dymally School of Nursing (MMDSON), Charles R. Drew University of Medicine and Science (CDU), Los Angeles, CA 90059, USA; 2Department of Family Medicine, Charles R. Drew University of Medicine and Science (CDU), Los Angeles, CA 90059, USA; 3Department of Internal Medicine, Charles R. Drew University of Medicine and Science (CDU), Los Angeles, CA 90059, USA; 4Physician Assistant Program, Charles R. Drew University of Medicine and Science (CDU), Los Angeles, CA 90059, USA; 5Office of Research, Charles R. Drew University of Medicine and Science (CDU), Los Angeles, CA 90059, USA; 6Department of Emergency Medicine, Mellie’s Bank Hospital, Tehran 1135933763, Iran; 7Department of Family Medicine, University of California Los Angeles (UCLA), Los Angeles, CA 90059, USA

**Keywords:** ethnic groups, COVID-19, telehealth service utilization, access to cellular network and Wi-Fi, older adults, African Americans

## Abstract

Background: The COVID-19 pandemic transformed healthcare delivery with the expansive use of telemedicine. However, health disparities may result from lower adoption of telehealth among African Americans. This study examined how under-resourced, older African Americans with chronic illnesses use telehealth, including related sociodemographic and COVID-19 factors. Methods: Using a cross-sectional design, 150 middle-aged and older African Americans were recruited from faith-based centers from March 2021 to August 2022. Data collected included sociodemographics, comorbidities, technological device ownership, internet usage, and attitudes toward COVID-19 disease and vaccination. Descriptive statistics and multivariable regression models were conducted to identify factors associated with telehealth use. Results: Of the 150 participants, 32% had not used telehealth since the COVID-19 pandemic, with 75% reporting no home internet access and 38% having no cellular/internet network on their mobile device. Age, access to a cellular network on a mobile device, and wireless internet at home were significantly associated with the utilization of telehealth care. Higher anxiety and stress with an increased perceived threat of COVID-19 and positive attitudes toward COVID-19 vaccination were associated with telehealth utilization. Discussion: Access and integration of telehealth services were highlighted as challenges for this population of African Americans. To reduce disparities, expansion of subsidized wireless internet access in marginalized communities is necessitated. Education outreach and training by healthcare systems and community health workers to improve uptake of telehealth currently and post-COVID-19 should be considered.

## 1. Background

Due to the global spread of COVID-19, most healthcare clinics, facilities, and independent providers were forced to cease all in-person contact with patients, which meant cancellation of wellness visits, health screenings, elective surgeries, and other forms of care. The U.S. healthcare system had to radically shift its primary form of delivery to ensure that optimal care was given and COVID-19 prevention practices were adhered to, prompting the embrace of telehealth. Federal and state policies for Medicaid and Medicare were introduced to allow greater use of telehealth during the COVID-19 pandemic, which aimed to reduce exposure and transmission of the virus for both patients and providers [1,2]. In response to the exceptional need for remote medical monitoring and consultations during the COVID-19 crisis, telehealth experienced explosive growth in its utilization among healthcare institutions and providers. Even with the improvement in telehealth offerings and increase in usage, gaps in access and use still exist [3,4]. Several studies have shown that access to telehealth was not equitable across different population subgroups [5,6,7,8,9]. Moreover, a recent systematic review of published studies revealed that there are looming disparities in telehealth use by race, age, income, and other factors such as telehealth literacy, access to an electronic digital device, and internet connectivity [1].

Knowing that the vast majority of victims of COVID-19 were older adults and individuals with underlying chronic conditions [10,11], promoting beneficial use of telehealth among these two populations should remain a major focus of telehealth utilization research. The utilization of telehealth services among older adults also substantially increased during the outbreak [12]. Yet, older adults may face a lack of knowledge concerning their access and a lack of knowledge of digital technology, resulting in multiple disadvantages [13,14,15]. Pre-pandemic and current healthcare utilization data show disparities in telehealth use by race and ethnicity among older adults [13,16]. Lower rates of technology usage among older minority adults and lower socioeconomic groups have been extensively established, attributing to a digital divide among these groups [17]. Greater emphasis should center on telehealth usage among minority middle-aged and older adults, as they have a higher prevalence of chronic medical conditions, including hypertension, congestive heart failure, diabetes, chronic obstructive pulmonary disease (COPD) and asthma, and obesity, which are all risk factors for poor COVID-19 outcomes [18,19,20]. An examination of the Behavioral Risk Factor Surveillance System dataset from January 1999 through December 2018 showed extensive disparities among middle-aged and older African American adults and White adults, with regard to increased rates of diabetes, hypertension, and asthma [21].

Recent studies highlight the importance of telehealth in increasing access to care and promoting health equity for underserved and under-resourced populations [22]. However, the rapid adoption of telehealth may exacerbate existing inequities for African Americans, as individuals with lower incomes and older in age face more challenges when attempting to use telehealth services [17]. Healthcare access and quality are identified as major contributing factors to the increased COVID-19 burden among older African Americans [23]. Delay in care and poor access to specialized care during the pandemic was associated with an increase in severely uncontrolled chronic diseases, such as diabetes, particularly in vulnerable populations [24]. Yet, little knowledge exists about the health care usage or interruption in care during the COVID-19 pandemic among older African Americans [25,26,27,28]. It is in this context that addressing factors that caused a telehealth avoidance in accessing healthcare among middle-aged and older African Americans with multiple chronic diseases during the COVID-19 pandemic remains a major public health concern [29]. Indeed, interventional studies are urgently needed to promote appropriate use of telehealth among middle-aged and older African Americans with multiple chronic diseases [30,31,32,33].

### Aims

In response to the increased need for designing effective interventional strategies to promote appropriate telehealth utilization among African Americans, this study examined the frequency and correlates of telehealth utilization since the onset of the COVID-19 pandemic among a sample of underserved African Americans aged 55 years and older residing in the South Los Angeles, California. 

## 2. Materials and Methods

### 2.1. Design and Setting

This cross-sectional study was conducted in faith-based organizations (i.e., churches) in South Los Angeles. Eligibility was based on African American ethnicity, age 55 or more, and the presence of at least one chronic condition. Data was collected between March 2021 and August 2022. This research is a part of a larger study designed to assess the impact of a community-based, participatory, faith-based, multidisciplinary, theoretical-based intervention to reduce the risk of COVID-19 and enhance chronic care management among underserved middle-aged and older African American adults. Within the geographic area of South Los Angeles, African Americans constitute nearly 30% of the over 1 million residents [34].

### 2.2. Institutional Review Board (IRB)

The study protocol was approved by the IRB of the Charles R. Drew University of Medicine and Science (CDU), Los Angeles (IRB#: 1663247-1). All participants signed a written informed consent before being enrolled in this study. Participants received financial compensation.

### 2.3. Recruitment and Sampling

Non-random sampling was used for recruitment. A total number of 150 individuals were enrolled. All participants provided signed consent. The Charles R. Drew University of Medicine and Science approved the study.

### 2.4. Measurement 

#### 2.4.1. Demographics Characteristics

Various factors were assessed, including age, gender, educational attainment, and marital status as covariates. Educational attainment was operationalized as a continuous variable (number of years for school attendance). Higher scores indicated more years of education. In addition, we asked our participants whether they were married or lived with a partner, which was analyzed categorically as either married/lived with a partner or not married/did not live with a partner.

#### 2.4.2. Financial Strain

This variable was measured using four items. Participants were asked, “In the past 12 months, how frequently were you unable to: (1) buy the amount of food your family should have, (2) buy the clothes you feel your family should have, (3) pay your rent or mortgage, and (4) pay your monthly bills”. Items were on a five-level response scale ranging from 1 (never) to 5 (always). A total “financial strain” score was calculated, with an average score of four items, ranging from 1 to 5. A high score was indicative of greater financial difficulty. These items are consistent with Pearlin’s list of low SES individuals’ chronic financial difficulties (Cronbach alpha = 0.92) [35].

#### 2.4.3. Number of Major Chronic Diseases

We asked participants to report whether they had been diagnosed with the following diseases: hypertension, diabetes mellitus, cardiovascular disease, COPD or asthma, and kidney disease. This variable had a potential range of 0 to 5, with a higher number indicating multi-morbidity.

#### 2.4.4. COVID-Related Constructs

We asked participants about COVID-19 vaccination history, COVID-19 diagnosis history, COVID-19 perceived threat, and COVID-19 knowledge, using multiple items borrowed from the PhenX COVID-19 library (https://www.phenxtoolkit.org/, accessed on 11 March 2021). The PhenX (consensus measures for Phenotypes and eXposures) Toolkit is a publicly available, web-based catalog of recommended, well-established measurement protocols of phenotypes and exposures [36,37].

#### 2.4.5. Attitudes toward COVID-19 Vaccination

This variable was measured using 10 items adopted from the Sage Vaccine Hesitancy Report. Participants were asked to report whether they strongly disagree, disagree, neither disagree nor agree, agree, or strongly agree with statements that were indicative of their attitudes toward the COVID-19 vaccination. Depending on their perception of telemedicine, these attitudes may be either positive if they agree with COVID-19 vaccination or negative if participants disagree with it. A total score was calculated, with an average score of 10 items, ranging from 1: very negative to 5: very positive. 

#### 2.4.6. Perceived Threat of Infection (Risk) of COVID-19

This variable was measured using five items adopted from the Social Psychological Measurement of COVID-19 (Rad-X). Using a scale of 0 to 7, participants were asked to report: (1) thinking about the coronavirus (COVID-19) makes me feel threatened; (2) I am afraid of the coronavirus (COVID-19); (3) I am worried that I or people I love will get sick from the coronavirus (COVID-19); (4) I am stressed around other people because I worry I will catch the coronavirus (COVID-19); and (5) I have tried hard to avoid other people because I do not want to get sick. A total “Perceived Threat” score was calculated, with an average score of five items ranging from 0 to 7. A high score was indicative of a greater perceived threat of infection.

#### 2.4.7. Knowledge of COVID-19

Twenty-one items that were publicized by Rad-X were used to assess the level of COVID-19 related knowledge of participants. Participants were asked to identify the symptoms of COVID-19, how the virus is spread, as well as how people can protect themselves from getting infected. A total score was calculated, with an average score of 21 items (true or false), ranging from 1 to 2. A high score was indicative of greater COVID-19 knowledge.

#### 2.4.8. Telehealth-Related Utilization

We used six items to capture telehealth-related medical care utilization. Participants were asked to report: (1–3) Have you participated in telehealth visits since the coronavirus pandemic? If yes, did you use the telephone or video for a visit(s); (4) do you ever go on-line to access the internet or to send and receive email; (5) when you use the internet, do you access it through a cellular network on your phone, a wireless network–Wi-Fi–at home, etc.; (6) how often do you access the internet (using a regular dial-up telephone, do you ever go online).

### 2.5. Data Analysis

Our analysis had three parts. The first section was a descriptive analysis of all participants. This descriptive work reported means and standard deviation for continuous measures and frequency and percentages for the categorical variables. Next, Pearson correlation was utilized to examine the bivariate association between all study variables, including socio-demographics, COVID-19-related measures, health, and the outcome variable (telehealth use since COVID-19). For the multivariable analysis, binary logistic regressions were run to examine the independent association of telehealth care utilization and all our exploratory variables. Both categorical and continuous independent variables are included in the model, regardless of the bivariate analysis results. For Binary Logistic Regression, Exp (B) the exponentiation of the B coefficient, which is an odds ratio and a 95% confidence interval, are reported. For the multivariate analysis, *p* values of less than 0.05 were considered significant. 

## 3. Results

### 3.1. Descriptive Analysis

Table 1 reports the characteristics of the study sample. This study included 150 African American individuals who were between the ages of 55 and 91 years (mean = 68.5 ± 8.66). Approximately 32% of participants were 75 years of age or older, with only 39% self-reporting as being married. Thirteen percent of participants never completed high school, and another 27% reported completing high school. Regarding health status, we noted the following health diseases/illnesses: hypertension (56%), diabetes mellitus (22%), COPD or asthma (22%), and cardiovascular-related diseases (11%). Over half of our participants (57%) rated their mental health as very good or excellent. Only thirty percent (30%) of participants self-reported their physical health as very good or excellent. Almost 28% reported their oral health as poor or fair.

Nearly one-third of participants (32%) had not used telehealth since the onset of the COVID-19 pandemic (Table 1). Only 19% and 49% of participants used video calls and telephone calls, respectively. Over 40% of participants who used video telehealth indicated that someone assisted them in joining the video call. However, both sub-groups indicated that they need additional technical support to ensure that future telehealth visits are effective. In addition, 30% of telehealth users were concerned that their provider could not physically examine them during telehealth. Additionally, 20% admitted that their medical questions or issues were not addressed (Table 1). 

At least two-thirds (66%) of telehealth users reported that the various parts of their telehealth visit (scheduling, the patient portal, video/phone specifications) were not well integrated. However, 73% of them expressed that they were satisfied (29%) or very satisfied (44%) with their most recent experiences with their telehealth visits. Furthermore, 62% felt that telehealth is a convenient form of healthcare delivery for them, with 56% having a stronger preference for telehealth compared to an in-person visit. Finally, when participants were asked, “what would make your experience with telehealth better?”, almost one-third (31%) pointed to having better access to high-speed/quality data plans, internet services, and technical support.

Technological access differed, with 40% reporting daily access to a computer compared to 19% only having access occasionally. Additionally, 41% indicated that they never go online to access the internet or to send and receive digital media or emails. Regarding wireless connectivity, 25% indicated that they have no internet at home, and 38% had no cellular network on their cellphone.

### 3.2. Bivariate Analysis

Table 2 shows bivariate correlations between telehealth utilization and other study variables. Findings revealed that age, gender, and education were not correlated with telehealth utilization. However, a number of chronic diseases (r = 0.252; *p* < 0.001), perceived risk of COVID-19 (r = 0.166; *p* < 0.05), and attitudes toward COVID-19 vaccination (r = 0.282; *p* < 0.001) were associated with telehealth utilization. In addition, having internet at home (r = 0.236; *p* < 0.001) and a cellular network (r = 0.207; *p* < 0.005) were significantly associated with telehealth utilization.

### 3.3. Multivariable Analysis

Table 3 presents the summary of the multivariable analysis using Poisson log-linear regression. According to this table, controlling for all other variables, older age (OR: 1.13; 95% CI: 1.03–1.23) and not being enrolled in Medicare (OR: 0.19; 95% CI: 0.05–0.67) were associated with higher utilization of telehealth. In addition, having the internet at home (OR: 11.3; 95% CI: 2.81–45.39) and having a cellular network (OR: 4.22; 95% CI: 1.21–14.77) were significantly associated with the utilization of telehealth. Having a positive attitude toward COVID-19 vaccination and a higher level of perceived risk of COVID-19 infection were also associated with telehealth care utilization. However, knowledge of COVID-19 symptoms and protection measures was not associated with telehealth care utilization.

## 4. Discussion

The findings of this study revealed that increased perceived risk of COVID-19 and positive attitudes toward COVID-19 vaccination were associated with telehealth utilization. In a sample of middle-aged and older African Americans with chronic health issues, our results showed participants who were not Medicare beneficiaries, had home internet, possessed a cellular phone, and were chronologically older were more likely to utilize telehealth services. Combined, these findings advance the literature for emphasizing the promising role of relevant culturally sensitive, theory-based telehealth services and interventions as a tool to improve the health status of an under-resourced minority population.

Our findings revealed a significant relationship between older age and higher utilization of telehealth among this under-rescourced population. Elam and colleagues (2022) uncovered that older adults in the US were less likely to have telehealth visits than in-person visits during the height of the COVID-19 pandemic [38]. However, older adults are likely to be interested in participating in a first-time telehealth visit [39], which should prompt healthcare systems to offer more assistance and guidance for telehealth to the older population. Additional multiple lines of data suggest that COVID-19 was a particular threat to older adults with numerous comorbidities [40,41], which may have contributed to their interest in telehealth services. It is important to note that 40% required assistance to join telehealth services, which should prompt providers to ensure that this population has comfort with telehealth services or provide technological assistance. Furthermore, undesired COVID-19 outcomes such as hospitalization are highest among older adults with underlying diseases [25], as COVID-19-related hospitalization and death rates increase with age [25]. Therefore, efforts to engage older adults in telehealth practices may be beneficial to increase their engagement.

This study showed that positive attitudes toward COVID-19 vaccination and higher levels of perceived risk of a COVID-19 infection were associated with telehealth care utilization. It has been evidenced that having a greater degree of understanding of COVID-19 was connected with having more favorable views toward preventive actions, such as using a mask and adhering to social distancing measures. Compared to non-Hispanic Whites, African Americans were at an increased risk of experiencing severe COVID-19 outcomes [42,43]. In addition, the increased incidence of severe COVID-19 among African Americans is partly due to higher infection rates, which suggests that COVID-19 disparities most likely result from increased vulnerability [42,43]. Thus, those who may have perceived themselves at risk may have been more likely to communicate with a healthcare provider.

Medical insurers have noted that there was an increase in telehealth billing claims for residential areas that are predominantly African American [2]. Our results supported this claim, with 19% reporting a video call with a provider, compared to 49% who had a telephonic visit with their provider. Our findings show that those who had strong attitudes toward protecting themselves against a COVID-19 infection were more likely to continue with care visits with their provider. However, 30% reported no usage of telehealth. Therefore, it should be a priority that providers continue to assess attitudes toward COVID-19 and other major chronic illnesses, and collective efforts should be made to increase positive messaging about COVID-19 and health status, especially during all telehealth and in-person care visits.

Our results also revealed that having internet at home and having a cellular network were significantly associated with the utilization of telehealth. A recent Pew Research Center report revealed the widening of the national digital divide, as those with lower incomes have lower rates of ownership of a smartphone, home internet, or desktop/laptop/tablet computer, resulting in decreased technological access [44]. In our entire study population, 25% indicated that they have no internet at home, and 38% had no cellular network on their cellphone. One of the major obstacles to the widespread use of telehealth is the influence of socioeconomic status, resulting in lower access to and proficiency with various forms of technology, especially for impoverished populations. Some contributing factors include financial hardship, illiteracy, a lack of enthusiasm or desire, and an absence of necessary resources to make use of available technologies [45,46]. A recent study found that compared to their White counterparts, African Americans were less likely to participate in telehealth and telemedicine visits [8]. The COVID-19 pandemic led to healthcare providers shifting to online and remote modalities for clinical care, as opposed to in-person visits [47]. 

Additionally, we documented that not being enrolled in Medicare was also associated with telehealth use. Webber and colleagues found that individuals with commercial or private health insurance were likelier to have appropriate technological devices for telehealth use compared to Medicare patients with a prepaid phone or no phone at all [48]. Further studies support our findings that commercial insurance holders utilize telehealth services more frequently than Medicare beneficiaries [49]. During the COVID-19 pandemic, the Centers for Medicare and Medicaid Services approved telehealth waivers as a component of the Public Health Emergency framework, which likely increased the utilization of telehealth services by Medicare beneficiaries [50]. Even though the offering of telehealth services among Medicare beneficiaries increased from 18% to 63%, pre-pandemic to November 2020, minorities and older adults had less availability of telehealth equipment. Additional studies reported lower adoption of telehealth services among African American and Latino Medicare beneficiaries [51].

Our data also showed a relationship between the total number of diagnosed chronic diseases and telehealth utilization at the bivariate level. With only 30% of the study population rating their physical health as very good or excellent, participants may have sought medical care via telehealth methods for their chronic health issues and quality of medical life. Ng and Park (2021) found that individuals with multiple comorbidities were likely to have telehealth services offered by their primary care provider compared to an in-person visit [52]. Supporting evidence has confirmed our findings that individuals with an increasing number of comorbidities are likely to be offered and use telehealth services [53]. Moreover, telehealth interventions have benefited African Americans for various diseases, including diabetes and glycemic control [54], gastrointestinal diseases [55], and anxiety and depression [56]. Social isolation, combined with other risk factors such as depression, may contribute to worsening disease management for these individuals with chronic diseases [40]. Due to the consistent messaging of practicing social distancing, an increase in telehealth utilization and treatments may be forecasted as a preferred method for the management of chronic diseases. However, those with multiple comorbidities and psychosocial needs may struggle to engage in telehealth visits and face difficulty navigating services [57]. It is crucial to study the effectiveness of treatments to know which chronic diseases may be managed remotely and which need personal contact.

It is important to highlight that over two-thirds of the study population believed that the processes of scheduling and participating in a telehealth visit were not well integrated. A study that explored telehealth perspectives among under-resourced communities in the South Los Angeles area found that African Americans were more concerned regarding the lack of privacy and confidentiality and the physical absence of the provider, compared to Latinos [58]. This may contribute to our finding of 20% reporting that their questions were not answered during telehealth visits. Wegerman and colleagues (2021) propose that healthcare systems must assess and ensure that there is full support for telehealth with both internal and external partners prior to implementation for integration ease [59]. Various healthcare providers, including physicians and nurses, have exposed telehealth disparities in areas such as access to care, bandwidth connectivity, availability of devices to perform telehealth, and socioeconomic and language barriers [60]. Moreover, they discuss that smartphone interfaces and funding for patient advocacy is essential for the success of telehealth.

### 4.1. Implications for Programs, Practice, and Research

These findings have significant implications for issues such as the introduction of telehealth, patient satisfaction, and the interaction between healthcare practitioners and patients. In addition, telehealth can benefit under-resourced populations, including those experiencing transportation challenges or other socioeconomic disparities. Meeting the needs of African American older adults during the COVID-19 pandemic may show innovation that can be translational for local governments and traditional safety net providers within the social work milieu [61]. Moreover, access can be increased for these populations to have health encounters with providers of different specialties, as waiting times are likely to decrease.

To increase health equity, health systems and providers must develop and continuously improve infrastructure for greater access and availability of telehealth for all individuals, which includes compatibility of various technological devices for audio and video visits. With the increased usage of telehealth, many providers could maintain their clinical priorities while maintaining social distance from their patients during the height of the pandemic. As projected, telehealth will continue to rise significantly following the COVID-19 pandemic, necessitating expanded access and accessibility, especially for under-resourced minorities. Additionally, these telehealth visits should incorporate a multidisciplinary approach, in which various providers can meet with the patient in a single visit, such as the physician, pharmacist, and social worker. 

Healthcare policies centered on incentivizing telehealth adoption for providers and patients should address the inequalities in mobile technology among under-resourced minorities. Technological devices owned by lower-income individuals, such as prepaid cellphones, may have fewer features and poorer capability to access telehealth components, such as video call visits or reviewing their online health records. Moreover, these populations may reside in communities with environmental and structural inequalities, with little or no wireless internet services, also known as “WiFi deserts”, primarily required for technological device use. Future policies should increase internet access, broad infrastructure, and available intelligent devices that will economically and technologically benefit under-resourced populations.

When establishing strategies to promote and implement telehealth in urban, underserved African American populations, it will be crucial to examine their attitudes and receptiveness toward the healthcare innovation of telehealth. The COVID-19 pandemic catalyzed fundamental shifts in healthcare delivery, leading to rapid integration in facilities frequented by marginalized populations and exacerbating health disparities [61]. Therefore, certain population groups, including our population of older African Americans and Latino groups, may have yet to receive in-person guidance on how to use telehealth services, learning on their own during the pandemic. This likely precipitated barriers toward telehealth, including specific patient factors, such as time inconvenience, cognitive or sensory impairment, and lack of perceived benefit. However, this may be compounded by other chronic health issues faced by African American and Latino groups in South Los Angeles, including low back pain [62], poor nutritional status [63], and frequent emergency care utilization [64]. Future research should explore racial or ethnic variations among patients who used various telehealth options during the COVID-19 outbreak. In addition, literature has found that racial/ethnic minorities and older adults are less likely to participate in telehealth [65,66,67,68]. As African American and Latino groups are underrepresented in studies surrounding telehealth, implementing interventions to improve clinical outcomes with telehealth care delivery methods may be effective.

Due to the high vulnerability of this population, the reduction of COVID-19 risk and the management of chronic diseases in older African Americans with a propensity for multiple comorbidities require urgent attention. Our national healthcare system was unprepared with the onset of the COVID-19 pandemic, which worsened the health status of under-resourced communities. As rates of COVID-19 are decreasing, providers are still managing the long-term effects of COVID-19, coupled with care management of other major chronic diseases. To combat future crises of new infectious diseases, innovative interventions to increase telehealth use among this population is critically warranted. As health-based technological systems and interfaces continue to advance rapidly, certain populations may not gain access to these innovations and instead face worsening health effects. Current telehealth methods, such as emailing/texting providers or sharing results from wearable technology are not utilized by all individuals, increasing disparities. Efforts to increase telehealth utilization among older African Americans with multiple comorbidities in the context of COVID-19 are warranted to improve health access and status.

### 4.2. Limitations

This study had several limitations. First, this study had a small sample size, which limits our confidence regarding the generalizability of findings, particularly negative associations. A second limitation was the study’s cross-sectional design. All our study variables were measured at an individual level, and we did not collect neighborhood-level data or data from the healthcare system. We also did not measure mistrust in the healthcare system nor any stigma associated with it. Also, this study did not assess the frequency of in-person visits with providers or availability of various types of delivery methods offered by their provider (e.g., in-person, telehealth, home visit). Moreover, we also did not differentiate between different chronic diseases. While we pooled all chronic diseases together, some conditions may correlate differently to telehealth utilization. Additionally, participant enrollment and data collection period lasted from March 2021 to August 2022, which occurred during various waves and surging strains of the COVID-19 pandemic. However, the COVID-19 vaccination had been nationally available since the inception of the study, and our study reflects attitudes and behaviors among an under-resourced group who has been negatively impacted overall with severe COVID-19 outcomes. According to the findings of this study, African Americans who live in different regions of the United States of America have varying rates of telehealth utilization in their respective regions. The findings of Adepoju and colleagues (2022) revealed minority groups like African Americans and Hispanics who lived further away from healthcare facilities found telehealth very beneficial [69]. However, our study focused on older African Americans, who reside in one of the under-resourced urban communities in the nation and face multiple other socioeconomic and health disparities [70,71,72,73]. Moreover, the high expense of technology, inadequate internet connectivity, poor accessibility for the impaired, and the prevalence of low-performing gadgets all contribute to a general lack of access to technology.

## 5. Conclusions

Risk reduction of COVID-19 and chronic disease management for African American older adults with multiple comorbidities are highlighted as difficult challenges due to the high vulnerability of this group, a fact that warrants critical attention [74,75,76,77,78]. Proactive efforts to increase telehealth utilization associated with multiple comorbidities among older African Americans amid COVID-19 are warranted, as they can increase health access. In addition, providers and healthcare systems performing wide-scale implementation of telehealth services should address and resolve barriers faced by under-resourced minority populations. Education outreach and training by healthcare systems and community health workers to improve uptake of telehealth currently and post-COVID-19 should be considered. Finally, to reduce disparities, expansion of subsidized wireless internet access in marginalized communities is necessitated.

## Figures and Tables

**Table 1 ijerph-20-02675-t001:** Demographic, financial, and health-related characteristics and telehealth use among a sample of underserved, community-based, middle-aged and older African American adults (n = 150).

	N (%)
Gender Male Female	45 (30) 105 (70)
Age 55–64 65–74 75 and older	48 (32) 68 (45) 34 (23)
Education No High School Diploma High School Diploma Some College/Graduate	19 (13) 40 (27) 91 (60)
Medicare No Yes	78 (52) 72 (48)
Telehealth use since COVID-19 No Yes	47 (32) 102 (68)
Type of Telehealth Telephone Video Not Used	73 (49) 29 (19) 47 (32)
Cellular Network (Phone) No Yes	52 (38) 85 (62)
Internet (Wi-Fi) at home No Yes	35 (25) 106 (75)
	Mean ± SD
Age	68.5 ± 8.66
Financial Strains (1 = never to 4 = always) Food Rent/Mortgage Clothes Utilities	1.72 ± 1.03 1.64 ± 1.06 1.77 ± 1.20 1.73 ± 1.15 1.75 ± 1.11
Number of Major Chronic Conditions	1.22 ± 1.01
COVID-19 Knowledge (0: low to 2: high)	1.52 ± 0.37
COVID-19 Perceived Threat and Stress (0: no stress to 7: significant stress)	3.56 ± 2.21
COVID-19 Vaccination Attitude (1: very negative to 5: very positive)	4.03 ± 0.74

**Table 2 ijerph-20-02675-t002:** Bivariate correlations between telehealth utilization and other study variables.

	1	2	3	4	5	6	7	8	9	10	11
Telehealth	--										
2. Gender	−0.099	--									
3. Age	0.097	−0.071	--								
4. Education	0.141	−0.017	−0.143	--							
5. Medicare	−0.095	−0.105	0.347 **	−0.050	--						
6. Financial Strain	−0.121	−0.014	−0.112	−0.243 **	−0.094	--					
7. Chronic Illness	0.252 **	0.007	0.181 *	−0.037	0.143	0.022	-				
8. Perceived COVID Risk	0.166 *	−0.090	0.048	−0.021	−0.022	0.034	0.162	--			
9. Knowledge of COVID	0.138	−0.060	−0.041	0.115	0.142	−0.191 *	0.147	−0.012	--		
10. COVID Attitude	0.282 **	−0.098	0.002	0.189 *	−0.028	−0.087	0.156	−0.014	0.103	--	
11. A Cellular Network	0.207 *	−0.007	−0.218 *	0.062	0.069	−0.175 *	−0.017	0.046	0.167	0.134	-
12. Internet (at home)	0.236 **	0.024	−0.209 *	0.294 **	0.013	−0.071	0.037	0.008	0.156	−0.026	0.229 **

Note: ** Correlation is significant at the 0.01 level (two-tailed), and * correlation is significant at the 0.05 level (two-tailed).

**Table 3 ijerph-20-02675-t003:** Binary Logistic Regression, EXP(B) the exponentiation of the B coefficient (odds ratio and 95% confidence interval (CI)).

Independent Variables	B	Wald	EXP(B)	95% CI EXP(B)	Sig.
Gender	−0.193	0.116	0.825	0.273–2.494	0.733
Age	0.121	7.274	1.129	1.034–1.233	0.007
Education	0.062	0.053	1.064	0.629–1.799	0.818
Medicare	−1.682	6.648	0.186	0.052–0.668	0.010
Financial Strain	0.111	0.134	1.117	0.617–2.023	0.714
Chronic Conditions	0.309	2.214	1.363	0.906–2.048	0.137
Internet (at home)	2.424	11.651	11.287	2.807–45.393	0.001
Cellular Network (on phone)	1.441	5.093	4.225	1.209–14.767	0.024
Perceived Threat of COVID-19	0.274	4.483	1.315	1.021–1.694	0.034
Knowledge of COVID-19	−0.588	0.728	0.556	0.144–2.144	0.394
Attitude Toward Vaccination	0.872	4.389	2.391	1.058–5.404	0.036
Constant	−13.027	8.091	0.000	N/A	0.004

−2 Log Likelihood = 95.9; Cox & Snell R Square = 0.288; Nagelkerke R Square = 0.421.

## Data Availability

The data sets used and analyzed in the current study are available from the corresponding author for collaborative studies. Personal identification details of the participants were separated from the completed questionnaires. The data were stored in a locked room at the Charles R. Drew University of Medicine and Science (CDU). No information relating to identifiable individuals was disseminated at all. The data sets used and analyzed in the current study are available from the corresponding author for collaborative studies. Code availability: N/A.

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
