# Peer review of "Factors Associated with Telehealth Utilization among Older African Americans in South Los Angeles during the COVID-19 Pandemic"

_ijerph, 2023, doi:10.3390/ijerph20032675_

Round 1

Reviewer 1 Report

Dear Authors,

 Thank you for the opportunity to review your work. Its purpose and results are very important and extremely interesting. I have only a few suggestions that may affect the quality of the article and its value to readers.

1. The abstract is too extensive - it is worth shortening it to the sizes suggested by the journal.

2. I suggest introducing the Introduction section with justification of the importance of the problem, presentation of the purpose of the work and its structure, as well as the contribution of your article to the selected field of science.

3. The literature review presented in the Background section is too general. I propose to divide it into sections in accordance with the measures presented in methods (financial strains, telehealth). A similar structure should be used in the Discussion section to better present the essence of your results.

4. In the results section, it is worth commenting on the results contained in the tables in more detail.

5. In the Discussion, it is worth referring your results more closely to the results of previous papers presented (or those that will be presented) in the literature review.

Good luck!

Author Response

Thank you for the opportunity to review your work. Its purpose and results are very important and extremely interesting. I have only a few suggestions that may affect the quality of the article and its value to readers.

Responses: Many thanks for very thoughtful comments and suggestions.

  1. The abstract is too extensive - it is worth shortening it to the sizes suggested by the journal.

Thank you for your comment. We have condensed the abstract to adhere to the journal guidelines.

  1. I suggest introducing the Introduction section with justification of the importance of the problem, presentation of the purpose of the work and its structure, as well as the contribution of your article to the selected field of science.

Thank you for your comment.  We have reformatted our introduction based on the strategies proposed.  Please see the edited introduction, which is listed on page 3.

  1. The literature review presented in the Background section is too general. I propose to divide it into sections in accordance with the measures presented in methods (financial strains, telehealth). A similar structure should be used in the Discussion section to better present the essence of your results.

Thank you for your comment.  We have added new references and included more focus on telehealth disparities for this population and for the healthcare system.  We have also implemented this similar focus in the Discussion section.  Please see changes for both introduction and discussion sections.

  1. In the results section, it is worth commenting on the results contained in the tables in more detail.

Thank you for this comment. We have added additional verbiage to describe the findings of the tables in the results section.  The verbiage is listed in results section 3.1 - 2nd paragraph, section 3.2. 1st paragraph, and section 3.3 1st paragraph.

  1. In the Discussion, it is worth referring your results more closely to the results of previous papers presented (or those that will be presented) in the literature review.

Thank you for this comment. We have included prior references in the introduction in the discussion.

Reviewer 2 Report

Reviewer’s observations 8th January 2023

 The authors are to be congratulated on doing a detailed study on a very important area – namely understanding factors influencing utilization of telehealth . The study has been done in a specific community in a focused geography  in a small number studied 150. Study restricted to South Los Angeles. The long duration of study March 2021 to August 2022 may itself contribute to observations differrnt if it was over a 3 month period.  Reader has no idea if the observations from this number, if extrapolated truly represents the population of “ older African Americans ” . Details of total no of older AA in that geography at that time is required. Title should clearly mention south LA .  Discussion and references pointing out that there are or there are no geographical variations in this group – i.e AA living in other parts of the USA -  is required. There is no information regarding availability or non availability of direct access to physical consults. Was teleconsultation used during that time because there was no alternative or as an optional  choice . Does the same situation prevail in 2023 as in 2021. Incidence of chronic diseases in The AA population including one CD if info is not available. This needs to be mentioned . The most important question that needs to be addressed is  so what ? – in other words how does the information provided by the authors relevant in the year 2023. If the study could be repeated in 2023 without the pandemic being what it was then it would be very useful and would considerably add to the value of the paper irrespective of the new findings . Since the focus is on AA it will be useful if similar studies in Hispanic whites in that geography at that time are discussed.   

Author Response

2nd Reviewer Comments and Suggestions

The authors are to be congratulated on doing a detailed study on a very important area – namely understanding factors influencing utilization of telehealth. The study has been done in a specific community in a focused geography in a small number studied 150. Study restricted to South Los Angeles.

Responses: Thank you very much for your comments and suggestions.

  1. The long duration of study March 2021 to August 2022 may itself contribute to observations different if it was over a 3 month period.  

Thank you for this comment. We identified this information in the Limitation section.

  1. Reader has no idea if the observations from this number, if extrapolated truly represents the population of “older African Americans”. Details of total no of older AA in that geography at that time is required.

Thank you for this comment.  To ensure the recentness of the data, we have included the total population of African Americans who reside in this geographical region. This data is now included in the Methods section 2.1 – 1st paragraph.

  1. Title should clearly mention south LA.  

Thank you for this comment.  We have now changed the title to include the location.

  1. Discussion and references pointing out that there are or there are no geographical variations in this group – i.e AA living in other parts of the USA -  is required.

Thank you for this comment. We identified this information in the Limitation section.

  1. There is no information regarding availability or non availability of direct access to physical consults. Was teleconsultation used during that time because there was no alternative or as an optional  choice. Does the same situation prevail in 2023 as in 2021.

Thank you for this comment. We added this information in the Limitation section.

  1. Incidence of chronic diseases in The AA population including one CD if info is not available. This needs to be mentioned.

Thank you for this comment. We identified this information in the Introduction section – 2nd paragraph.

  1. The most important question that needs to be addressed is  so what ? – in other words how does the information provided by the authors relevant in the year 2023. If the study could be repeated in 2023 without the pandemic being what it was then it would be very useful and would considerably add to the value of the paper irrespective of the new findings.

Thank you for this comment. We added this information in Section 4.1 – 5th paragraph.

  1. Since the focus is on AA it will be useful if similar studies in Hispanic whites in that geography at that time are discussed.   

Thank you for your comment. We have included similar studies for the Latino population in South Los Angeles in Section 4.1 – 4th paragraph.

Round 2

Reviewer 1 Report

Dear Authors,

thank you very much for your amendments.

I accept the paper in the current form.

Congratulations and good luck with future research.